# Spatiotemporal distribution and control of echinococcosis in Xinjiang, 2005–2023

Zhe Yin[1,2], Guangzhong Shi[1], Yalikun Maimaitiyiming[1], Qi Wang[1], Kaisaer Tuerxunjiang[1], Adili Simayi[1]*, Jiangshan Zhao 🄳[1]*

1 Xinjiang Uygur Autonomous Region Center for Disease Control and Prevention, Urumqi, China,
2 School of Public Health, Xinjiang Medical University, Urumqi, China

* zjscdc@163.com (JZ); adl1216@sina.com (AS)

## Abstract

### Objective

This study investigated the epidemiological characteristics of echinococcosis in Xinjiang, aiming to provide evidence for evaluating prevention and control progress and guiding future strategies.

### Methods

Reported echinococcosis incidence data from Xinjiang during 2005–2023 were obtained from the China Infectious Disease Surveillance and Reporting System. Temporal trends and spatial distributionwere analyzed using joinpoint regression and spatial autocorrelation methods. Additionally, a linear regression model (LR), spatial autoregressive model (SAR), and spatiotemporal autoregressive model (STAR) were constructed to quantitatively assess the impact of factors such as year, spatial lag, and temporal lag on incidence rates.

### Results

The reported incidence ranged from 1.95 per 100,000 in 2005 to 10.85 per 100,000 in 2017, followed by a gradual decline, though levels in 2023 (8.12 per 100,000) remained higher than those in 2005. Spatially, incidence exhibited significant positive spatial correlation (Moran's $I$: 0.097 in 2008 to 0.400 in 2010, all P-values < 0.001), with high-incidence counties concentrated in northern and central Xinjiang, while low-incidence counties were mainly located in the south. Northern Xinjiang, particularly areas centered around Tacheng Prefecture, Bortala Prefecture, and Ili Prefecture, was identified as a hotspot, whereas Kashgar and Hotan Prefectures in southern Xinjiang formed cold spots. Model analysis indicated that the STAR model provided the best fit (residuals range: 0.020 [−0.292, 0.317], AIC = 2,549.4), with year, spatial lag ($\beta = 0.323$), and temporal lag ($\beta = 0.420$) significantly affecting incidence

which permits unrestricted use, distribution, and reproduction in any medium, provided the original author and source are credited.

**Data availability statement:** All data necessary to replicate the study's key findings—including annual reported incidence, county-level median incidence , trend test results, spatial analysis metrics, and statistical model parameters—are available in the main text (Tables 1–6) and Supporting information files (S1 Table and S1 Fig). Full raw datasets cannot be publicly shared due to the confidentiality of county-level epidemiological data in Xinjiang and institutional data management policies (Xinjiang Uygur Autonomous Region Center for Disease Control and Prevention). Requests for non-public data should be directed to the Ethics Committee of the above institution via email: jbs@xjcdc.com, with a valid research proposal and institutional approval.

**Funding:** This work was supported by Xinjiang Uygur Autonomous Region Key Research and Development Project (2022B03013-1 to JZ). The funders had no role in study design, data collection and analysis, decision to publish, or preparation of the manuscript.

**Competing interests:** The authors have declared that no competing interests exist.

rates ($P < 0.05$). The findings suggest that echinococcosis incidence in Xinjiang exhibits temporal continuity and spatial spillover effects.

## Conclusion

Echinococcosis in Xinjiang exhibited marked spatiotemporal heterogeneity during 2005–2023. While national and regional prevention programs contributed to reducing incidence after 2017, the burden remains substantial. These findings underscore the need for sustained and regionally coordinated prevention and control strategies to prevent potential resurgence.

## Author summary

Echinococcosis is a severe zoonotic parasitic disease that remains highly endemic in Xinjiang, China, posing significant public health and economic burdens. Although comprehensive control measures have been implemented since 2005, the long-term epidemiological patterns and effectiveness of these interventions remain poorly understood. This study integrated surveillance data from 96 counties in Xinjiang between 2005 and 2023 and applied trend analysis, spatial autocorrelation, and spatiotemporal modeling to systematically evaluate the disease's spatial and temporal dynamics. Results indicated that, following the progressive implementation of national and regional control programs, the reported incidence peaked in 2017 and subsequently declined; however, pronounced spatial clustering persisted, with high-risk areas predominantly located in southern Xinjiang. Further analysis suggested that, while comprehensive control measures effectively reduced overall incidence, variations in intervention implementation led to uneven control outcomes across regions. This study provides systematic evidence on the long-term epidemiology and control effectiveness of echinococcosis in Xinjiang and offers scientific guidance for optimizing resource allocation, strengthening interventions in high-risk areas, and developing more targeted regional control strategies.

## Introduction

Echinococcosis is a zoonotic parasitic disease caused by the larval stage of *Echinococcus* tapeworms, with a widespread global distribution. Cases have been reported on every continent except Antarctica, and its impact is particularly severe in agricultural and pastoral regions, posing a significant threat to human health and economic development [1,2]. It is estimated that echinococcosis causes an annual economic loss of up to 3 billion USD [3]. Given its substantial burden, the World Health Organization (WHO) has designated echinococcosis as a priority disease for control [2].Western China is a major endemic region for echinococcosis, where the situation remains severe. Statistics indicate that approximately 170,000 individuals

are infected, with an estimated 50 million people at risk of contracting the disease [4]. Xinjiang Uygur Autonomous Region (hereafter referred to as Xinjiang), located in the northwestern frontier of China, has vast grasslands that provide favorable conditions for livestock farming, making it one of the country's key pastoral regions. Frequent human-livestock interactions in pastoral activities, combined with a unique ecological environment that favors the survival and reproduction of parasites and their hosts, have contributed to the widespread transmission of echinococcosis in Xinjiang [5].

Nationwide, Xinjiang is among the most severely affected provinces. Of the 370 echinococcosis-endemic counties in China, 81 are located in Xinjiang [6,7]. To address this high prevalence, the Chinese government and Xinjiang regional authorities have implemented a series of echinococcosis control programs since 2005. These measures include free treatment for patients, regular dog deworming campaigns, targeted population screening, health education initiatives, and improvements in pastoral sanitation. Such programs have been crucial in enhancing case detection and reducing transmission in endemic areas [8,9].This high prevalence poses significant challenges to both the health of local residents and the sustainable development of the livestock industry in Xinjiang. For affected individuals, echinococcosis can cause severe damage to vital organs such as the liver and lungs, leading to serious health complications, reduced quality of life, and increased medical burdens on households [10]. In livestock, echinococcosis infection impairs growth and reduces productivity, which may negatively affect the economic viability of the livestock industry [11]. Given the potential public health threat posed by echinococcosis in Xinjiang, a comprehensive understanding of its epidemiological dynamics is essential for improving prevention and control efforts. In this context, the present study systematically analyzed the spatiotemporal distribution of echinococcosis in Xinjiang from 2005 to 2023, aiming to characterize temporal trends and spatial clustering at the county level, summarize the implementation of major control measures during the study period, and evaluate the temporal and spatial associations between comprehensive control activities and disease incidence. The findings provide an evidence-based foundation for refining control strategies and optimizing public health resource allocation in endemic regions.

## Methods

### Data sources

Data on cases of echinococcosis in Xinjiang Uygur Autonomous Region and its 96 districts/counties during 2005–2023 were obtained from the China Infectious Disease Surveillance and Reporting System (IDSR), a national platform for mandatory infectious disease reporting. Cases were diagnosed according to the National Diagnostic Standard for Echinococcosis (WS 257–2006), which considers epidemiological history, clinical symptoms, imaging findings, and laboratory results. Only newly diagnosed cases were reported, and all case reports were verified through multi-level review by county, prefecture, regional, and national Centers for Disease Control and Prevention (CDC) to ensure accuracy and eliminate duplication. The incidence data were directly obtained from IDSR and were not independently calculated by the authors. Because echinococcosis is a chronic disease, the reported incidence reflects the number of confirmed and reported cases within a given period rather than the true rate of new infections. The base maps were sourced from the National Geographic Information Public Service Platform (www.tianditu.gov.cn), with the review number GS (2024) 0650. See S1 Table for county codes and median reported incidence.

**Overview of echinococcosis control and surveillance.** Echinococcosis in Xinjiang predominantly occurs in 81 counties (cities, districts). Since 2006, systematic control efforts have been implemented in endemic areas, focusing on comprehensive strategies including the control of infection sources, intermediate host management, population screening and treatment, and health education. By 2016, control measures had been implemented across all echinococcosis-endemic counties.

For human population protection, targeted groups in endemic areas—such as herders, slaughterhouse workers, and children—underwent regular ultrasound screening, and confirmed cases were promptly enrolled in standardized treatment programs. Health education was conducted through informational brochures, village-level lectures, and bilingual videos

to disseminate knowledge on prevention and enhance residents' self-protection awareness.For animal control, dogs were primarily subjected to a monthly deworming program, with excreta from treated animals handled safely and hygienically. Livestock control relied mainly on echinococcosis vaccination, with newborn lambs receiving two rounds of immunization to reduce intermediate host infection rates.

Regarding surveillance, from 2016 to 2018, echinococcosis monitoring was conducted in 39 counties. Beginning in 2019, the monitoring scope was expanded to cover all 81 endemic counties. In addition, since 2019, pilot comprehensive intervention programs have been implemented in three counties to explore more effective control strategies. It should be noted that all medical institutions across Xinjiang are required to report newly diagnosed cases in accordance with national regulations for Class C infectious diseases, regardless of whether the area is covered by control or surveillance programs. This integrated system of control and surveillance, which covers key hosts and high-risk populations and combines health education with case management, has provided systematic support for the reduction of echinococcosis incidence.

## Trend analysis

The Joinpoint regression analysis model automatically identifies trend change points and segments time series data to fit the optimal linear regression model, enabling an objective quantification of long-term disease trends. Compared with traditional regression methods, Joinpoint regression provides a more detailed and objective evaluation of disease trend characteristics. In this study, Joinpoint regression analysis was performed on the reported incidence of echinococcosis in Xinjiang from 2005 to 2023, calculating the Annual Percent Change (APC) and the Average Annual Percent Change (AAPC). If the number of joinpoints is greater than 0, APC is used to assess trends within each segment, while AAPC evaluates the overall average trend. If the number of joinpoints is 0, APC is used to assess the overall trend. At a significance level of $\alpha = 0.05$, an APC value less than 0 indicates a declining trend in reported incidence, while an APC value greater than 0 suggests an increasing trend. The absolute value of APC reflects the magnitude of the annual change.

The Mann-Kendall trend test is a non-parametric statistical method that is insensitive to outliers, making it suitable for small sample sizes or non-normally distributed data. In this study, the Mann-Kendall trend analysis was applied to the reported incidence of echinococcosis at the county level in Xinjiang. At a significance level of $\alpha = 0.05$, a Z-value less than 0 indicates a decreasing trend in reported incidence, whereas a Z-value greater than 0 suggests an increasing trend. The Theil-Sen slope estimator was used to quantify the trend rate. A positive slope with a larger absolute value indicates a higher annual growth rate, while a negative slope with a larger absolute value suggests a higher annual decline rate.

## Spatial correlation and heterogeneity analysis

The echinococcosis incidence database for each district and county was linked to the electronic administrative boundary map of Xinjiang. The mapping functionality of ArcGIS was utilized to generate thematic maps of reported incidence rates, providing a clear visualization of the geographical distribution characteristics.A global spatial autocorrelation analysis was conducted using Moran's I statistic to describe the spatial characteristics of reported incidence rates across the Xinjiang region. The formula is as follows:

$$I = \frac{\sum_{i=1}^{n} \sum_{j=1}^{n} w_{ij}(x_i - \overline{x})(x_j - \overline{x})}{S^2 \sum_{i=1}^{n} \sum_{j=1}^{n} w_{ij}}$$

(1)

Here, $n$ represents the number of districts and counties, $x_i$ denotes the reported incidence rate of echinococcosis in different districts and counties, $\overline{x}$ is the mean incidence rate, and $w$ represents the spatial weight. Moran's $I$ ranges from -1–1. At a significance level of $\alpha = 0.05$, a Moran's $I$ value greater than 0 indicates a positive spatial correlation, a value less than 0 suggests a negative spatial correlation, and a value of 0 signifies a random spatial distribution.

If a global spatial correlation is detected, a local spatial autocorrelation analysis is performed by calculating the local Moran's I to further elucidate the spatial autocorrelation patterns at the unit level. Local spatial autocorrelation analysis can identify four types of spatial clustering patterns: (1) "High-High clusters" and (2) "Low-Low clusters," which indicate similar incidence values among neighboring regions (i.e., areas with high incidence rates are surrounded by other high-incidence areas, and low-incidence areas cluster together); and (3) "High-Low clusters" and (4) "Low-High clusters," which indicate dissimilar values between a district and its neighboring regions, where high-incidence areas are surrounded by low-incidence areas or vice versa.

Hotspot and coldspot analysis was conducted using the Getis-Ord G statistic to identify spatial clusters of high and low echinococcosis incidence. The formula is as follows:

$$G_i = \sum_{j=1}^{n} w_{ij}x_j / \sum_{j=1}^{n} x_j \tag{2}$$

The parameters in this formula are the same as in Equation (1). At a significance level of α = 0.05, a $G_i$ value greater than 0 indicates a hotspot region for echinococcosis incidence, while a $G_i$ value less than 0 indicates a coldspot region.

### Model construction and evaluation

To ensure the completeness of the reported echinococcosis incidence data, the linear interpolation method was applied to impute potential missing values for each region. The Shapiro-Wilk test indicated that the reported incidence rates from 2005 to 2019 did not follow a normal distribution, with a skewness value of 2.513. To meet the assumptions required for subsequent modeling, the incidence data were log-transformed.

A linear regression (LR) model was constructed using the log-transformed echinococcosis incidence rate as the dependent variable, time (expressed in years) as the independent variable, and different districts and counties as factor variables. This model was primarily used to analyze the effect of temporal trends on incidence rates. To account for spatial autocorrelation, a spatial autoregressive (SAR) model was constructed. In addition to the variables included in the LR model, the SAR model incorporated spatial weights, allowing it to capture the influence of incidence rates in neighboring regions on the local incidence rate. To further explore the combined effects of spatial and temporal factors on echinococcosis incidence, a spatiotemporal autoregressive (STAR) model was constructed. This model incorporated both temporal and spatial lag terms. The temporal lag term captured the influence of past incidence rates on current incidence, while the spatial lag term reflected the impact of incidence rates in neighboring regions on the local incidence, enabling a comprehensive analysis of the interaction between spatial and temporal factors.

Parameter estimation for all three models was performed using the maximum likelihood estimation (MLE) method. Model fit was assessed using the Akaike Information Criterion (AIC), with lower AIC values indicating better model performance. Additionally, residual distribution characteristics, such as median residuals and interquartile ranges, were analyzed to ensure model reliability. The significance of model coefficients was assessed using the t-test at a significance level of α = 0.05, determining whether each factor had a significant impact on incidence rates.

## Results

### Descriptive spatiotemporal analysis

**Overall epidemiological characteristics.** From 2005 to 2023, a total of 23,934 cases of echinococcosis were reported in Xinjiang. The annual number of reported cases ranged from 377 to 2,573, with the annual incidence rate fluctuating between 1.95 per 100,000 and 10.85 per 100,000 (Table 1). Joinpoint regression analysis revealed a significant trend change point in 2017 (*P* < 0.05). From 2005 to 2017, the reported incidence rate showed a continuous upward trend, with an APC of 9.77% (95% CI: 6.30 - 14.24). From 2017 to 2023, the reported incidence rate significantly decreased,

**Table 1. Trends in reported incidence of echinococcosis in Xinjiang, China, 2005–2023.**

| year | number of cases | Reported incidence (1/100,000) | Growth rate of the number of cases | Growth rate of reported incidence |
|---|---|---|---|---|
| 2005 | 377 | 1.95 | – | – |
| 2006 | 752 | 3.82 | 99.47 | 95.90 |
| 2007 | 810 | 4.03 | 7.71 | 5.50 |
| 2008 | 910 | 4.42 | 12.35 | 9.68 |
| 2009 | 991 | 4.74 | 8.90 | 7.24 |
| 2010 | 1057 | 4.97 | 6.66 | 4.85 |
| 2011 | 1408 | 6.53 | 33.21 | 31.39 |
| 2012 | 1546 | 7.09 | 9.80 | 8.58 |
| 2013 | 1527 | 6.90 | -1.23 | -2.68 |
| 2014 | 1459 | 6.54 | -4.45 | -5.22 |
| 2015 | 1401 | 6.16 | -3.98 | -5.81 |
| 2016 | 1856 | 7.91 | 32.48 | 28.41 |
| 2017 | 2573 | 10.85 | 38.63 | 37.17 |
| 2018 | 1843 | 7.61 | -28.37 | -29.86 |
| 2019 | 1680 | 6.89 | -8.84 | -9.46 |
| 2020 | 1008 | 4.04 | -40.00 | -41.36 |
| 2021 | 927 | 3.65 | -8.04 | -9.65 |
| 2022 | 790 | 3.11 | -14.78 | -14.79 |
| 2023 | 1019 | 3.99 | 28.99 | 28.30 |

This table summarizes annual epidemiological data of echinococcosis, including year, number of reported cases, reported incidence (per 100,000 population), and year-on-year growth rates of cases and incidence (%). Dashes ("-") for 2005 indicate no baseline data for growth rate calculation. All data were extracted from the China Infectious Disease Surveillance and Reporting System (IDSR).

with an APC of -17.71% (95% CI: -27.81 to -10.48). The overall trend from 2005 to 2023 was not statistically significant ($P > 0.05$), with an AAPC of -0.28% (95% CI: -2.75 to 1.89), as shown in Fig 1.

**County-level epidemiological characteristics.** From 2005 to 2023, the median reported incidence rate at the county level in Xinjiang ranged from 0.79 to 3.69/100,000, with significant fluctuations observed. The highest reported incidence rate occurred in 2017, reaching 3.69/100,000, and then declined to 1.7/100,000 by 2023. There were notable differences in the incidence rates across counties. For example, in 2017, the highest reported incidence rate in a county reached 90.21/100,000, while the lowest was only 0.46/100,000, as shown in Table 2 and Fig 2.

The counties with the top five reported incidence rates each year exhibited dynamic changes and demonstrated a certain degree of regional clustering. In northern Xinjiang, counties such as Ulho, Mori, and Jeminay consistently ranked in the top five over multiple years. In 2005, Ulho's incidence rate was as high as 51.76/100,000, reflecting a high case detection. In 2008, Mori's incidence rate reached 85.23/100,000, the highest of that year. In 2017, Jeminay had a reported incidence rate of 90.21/100,000, maintaining a high level throughout that year and several previous years. Notably, some counties, such as Wenquan and Toli, showed improved rankings in later years, as shown in Table 3.

A trend test of the reported incidence rates of echinococcosis in the 96 counties of Xinjiang revealed that 17 counties, including Hejing, Bohu, and Tekes, exhibited a significant upward trend ($Z > 0$, $P < 0.05$). Among these, Hejing had the highest Theil-Sen slope (1.237, $P = 0.002$), showing the fastest growth rate. Geographically, 64.71% of the counties with increasing incidence rates were from southern Xinjiang (11/17), though their average slope of 0.181 was relatively low. In contrast, northern Xinjiang accounted for 35.29% of the counties (6/17), with a significantly higher average slope of 0.765, indicating a steeper growth trajectory. 11 counties, including Ulho, Mori, and Qitai, exhibited a significant downward trend ($Z < 0$, $P < 0.05$), with Ulho (-1.780, $P < 0.001$) and Mori (-1.180, $P = 0.025$) showing the steepest declines, as their

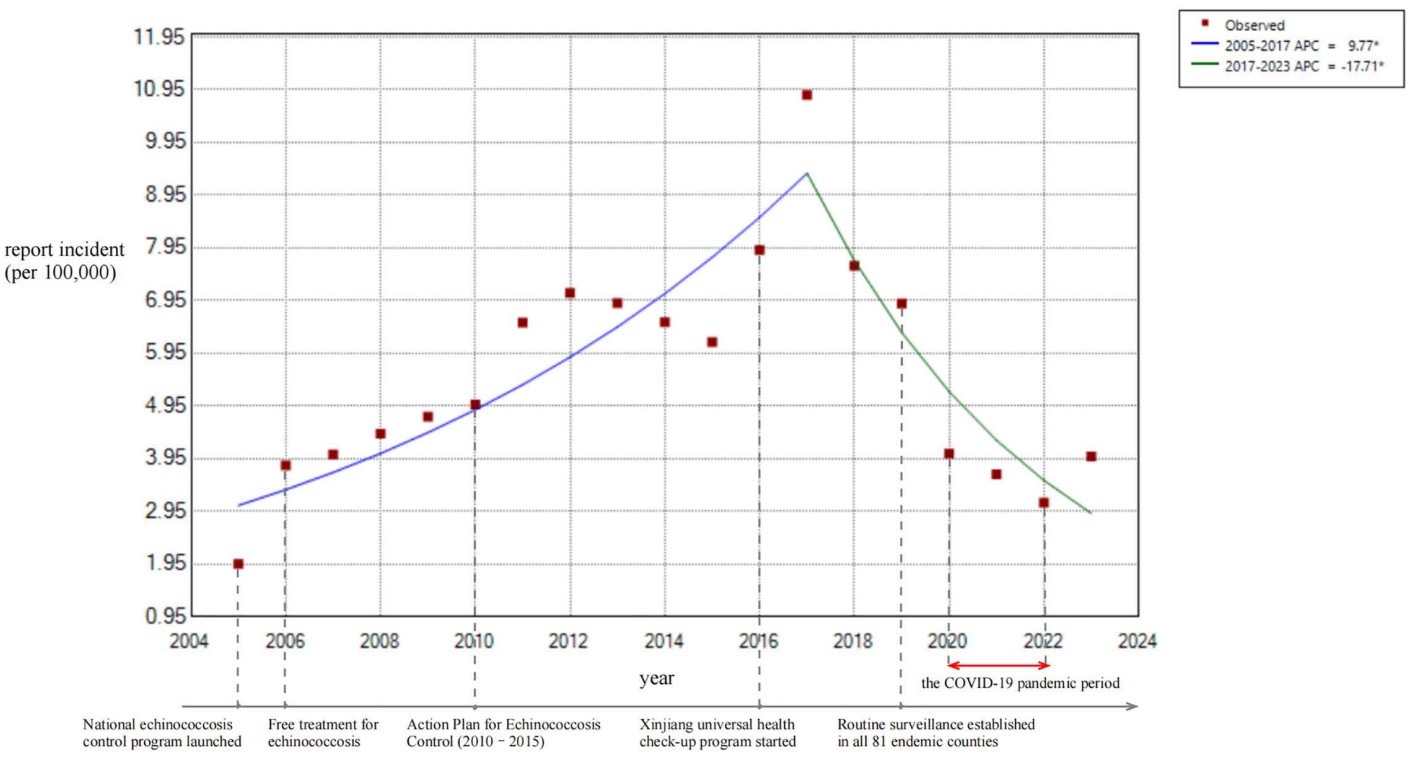

**Fig 1. Joinpoint regression analysis of reported incidence of echinococcosis in Xinjiang, China, 2005–2023.** This figure illustrates the joinpoint regression analysis of echinococcosis reported incidence (per 100,000 population) in Xinjiang over time. Red squares represent observed incidence data. The blue line depicts the trend from 2005 to 2017, with an annual percent change (APC) of 9.77% (P < 0.05). The green line depicts the trend from 2017 to 2023, with an APC of -17.71% (P < 0.05). Key public health events related to echinococcosis control in Xinjiang are annotated below the plot, including the launch of the national echinococcosis control program, free treatment initiatives, and the establishment of routine surveillance in all 81 endemic counties. Data were derived from the China Infectious Disease Surveillance and Reporting System (IDSR).

Theil-Sen slopes had absolute values greater than 1. All of the counties with decreasing trends were located in Karamay City, Urumqi City, and Changji Prefecture, as shown in Table 4.

## Spatial autocorrelation and heterogeneity analysis

The thematic map of echinococcosis reported incidence rates in Xinjiang's 96 counties from 2005 to 2023 shows that echinococcosis is present in most counties across the region, with significant spatial variation in the incidence rates. Low-incidence areas (0.15–5.00/100,000) are mainly concentrated in some southern counties, while moderate to high-incidence areas (5.01–100.00/100,000) are more prevalent in northern and central counties. Comparing different years, the geographic extent and intensity of echinococcosis fluctuated, with the affected area steadily expanding and the epidemic intensity peaking in 2017 before gradually decreasing, as shown in Fig 3. See S1 Fig for the median reported incidence of echinococcosis by county in Xinjiang.

The results of the spatial autocorrelation analysis are shown in Table 4. Global autocorrelation analysis revealed that the reported incidence rates over the 19 years exhibited positive spatial correlation, with global Moran's *I* values greater

**Table 2. Echinococcosis Reported Incidence at County-level in Xinjiang, 2005-2023.**

| Year | Reported Incidence | | Year | Reported Incidence | |
|---|---|---|---|---|---|
| | M(P25,P75) | Range | | M(P25,P75) | Range |
| 2005 | 1.86(0.79,3.81) | 0.24-51.76 | 2015 | 4.91(2.65,13.35) | 0.38-45.63 |
| 2006 | 3.46(1.37,8) | 0.15-34.25 | 2016 | 7.86(3.13,19.19) | 0.62-66.5 |
| 2007 | 5.15(1.54,7.59) | 0.24-30.63 | 2017 | 10.4(3.69,23.79) | 0.46-90.21 |
| 2008 | 4.04(1.25,9.23) | 0.14-85.23 | 2018 | 6.11(2.84,16.88) | 0.2-64.78 |
| 2009 | 5.78(1.72,10.07) | 0.23-35.78 | 2019 | 5.88(2.8,15.47) | 0.24-54.34 |
| 2010 | 5.65(1.81,11.31) | 0.22-54.92 | 2020 | 4.01(1.85,10.23) | 0.34-21.71 |
| 2011 | 4.95(2.02,13.73) | 0.41-45.65 | 2021 | 3.48(1.52,8.25) | 0.12-37.24 |
| 2012 | 7.2(2.17,16.6) | 0.22-59.11 | 2022 | 2.97(1.41,7.86) | 0.19-28.37 |
| 2013 | 5.23(2.75,14.74) | 0.39-51.36 | 2023 | 3.74(1.7,10.05) | 0.24-29.77 |
| 2014 | 5.74(2.64,13.63) | 0.38-49.23 | | | |

County-level reported incidence of echinococcosis in Xinjiang, China, 2005-2023.This table presents the annual county-level reported incidence of echinococcosis, with two statistical indicators: median (M) and interquartile range (P25, P75), and the full range of incidence. All incidence data are expressed as cases per 100,000 population. Data were extracted from the China Infectious Disease Surveillance and Reporting System (IDSR).

Annual Trends of Echinococcosis Reported Incidence in 96 Districts and Counties of Xinjiang , 2005-2023

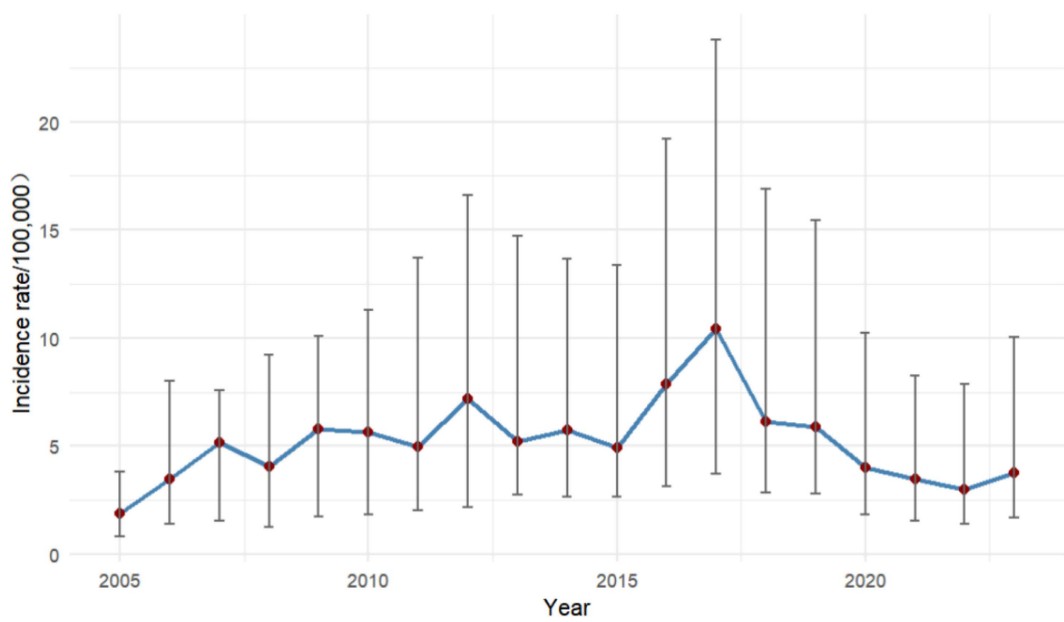

**Fig 2. Annual trends of echinococcosis reported incidence in 96 districts and counties of Xinjiang, 2005–2023.** This figure presents the annual trends of echinococcosis reported incidence (per 100,000 population) across 96 districts and counties in Xinjiang. The blue line and red dots represent the median incidence (with error bars indicating the interquartile range, P25 to P75). Data were extracted from the China Infectious Disease Surveillance and Reporting System (IDSR).

than 0 and *P*-values less than 0.001. The year 2010 showed the strongest clustering. Local spatial autocorrelation analysis indicated significant spatial heterogeneity in the reported incidence rates. The number of high-high (HH) clustering areas increased from 5 counties (Ulho, Toli, etc.) in 2005–18 counties (Tahcheng, Yining, etc.) in 2023. Low-low (LL) clustering areas were primarily concentrated in Hotan and Moyu in the Hetian Prefecture, as well as in Kashgar and Shule in

**Table 3. Top 5 Districts/Counties in Reported Incidence of Echinococcosis in Xinjiang, 2005–2023.**

| Year | Districts/Counties (Reported incidence) | | | | | Year | Districts/Counties (Reported incidence) | | | | |
|---|---|---|---|---|---|---|---|---|---|---|---|
| 2005 | Ulho (51.76) | Mori (20.28) | Jeminay (18.23) | Emin (14.47) | Barkol (14.01) | 2015 | Yumin (45.63) | Wenquan (44.16) | Jeminay (40.42) | Wuqia (31.63) | Zhaosu (29.99) |
| 2006 | Mori (34.25) | Wenquan (26.52) | Jeminay (20.58) | Hoboksar (19.32) | Qapqal (19.16) | 2016 | Wenquan (66.50) | Toli (61.66) | Jeminay (47.18) | Yumin (37.26) | Emin (36.54) |
| 2007 | Dasaka (30.63) | Mori (30.15) | Yumin (26.12) | Qapqal (21.52) | Qitai (21.33) | 2017 | Jeminay (90.21) | Zhaosu (83.45) | Wenquan (67.75) | Toli (66.09) | Habahe (53.75) |
| 2008 | Mori (85.23) | Jeminay (36.99) | Emin (32.3) | Akqi (27.94) | Yumin (21.91) | 2018 | Wuqia (64.78) | Toli (54.25) | Jeminay (40.5) | Emin (40.4) | Wenquan (37.84) |
| 2009 | Mori (35.78) | Jeminay (33.94) | Toli (31.8) | Yumin (30.51) | Tahcheng (30.01) | 2019 | Wuqia (54.34) | Toli (49.68) | Wenquan (34.26) | Hoboksar (32.74) | Qapqal (31.38) |
| 2010 | Yumin (54.92) | Toli (42.79) | Emin (33.24) | Hoboksar (30.51) | Jimsar (25.91) | 2020 | Zhaosu (21.71) | Emin (21.57) | Hoboksar (21.06) | Hejing (20.19) | Kalpin (20.13) |
| 2011 | Mori (45.65) | Wuqia (44.08) | Hoboksar (43.31) | Yumin (42.68) | Hexud (35.83) | 2021 | Yumin (37.24) | Tashkurgan (19.66) | Hejing (19.48) | Hoboksar (19.4) | Qapqal (18.15) |
| 2012 | Hoboksar (59.11) | Wuqia (56.03) | Zhaosu (53.77) | Ulho (50.9) | Toli (46.68) | 2022 | Wenquan (28.37) | Hoboksar (22.74) | Qapqal (22.29) | Hejing (18.35) | Zhaosu (17.07) |
| 2013 | Wenquan (51.36) | Zhaosu (49.28) | Jeminay (47.1) | Nilka (37.77) | Wuqia (36.86) | 2023 | Hejing (29.77) | Toli (26.19) | Qapqal (25.3) | Zhaosu (25) | Nilka (23.61) |
| 2014 | Jeminay (49.23) | Wuqia (48.18) | Yumin (46.38) | Hoboksar (39.28) | Wenquan (39.06) | | | | | | |

This table lists the five districts/counties with the highest annual reported incidence of echinococcosis. The values in parentheses following each district/county name represent its corresponding reported incidence, expressed as cases per 100,000 population. Data were extracted from the China Infectious Disease Surveillance and Reporting System (IDSR).

**Table 4. Trend Test Results of Echinococcosis Reported Incidence at County-level in Xinjiang.**

| Trend | Count | Z value | Districts/Counties | *P* value |
|---|---|---|---|---|
| significant upward trend | 17 | >0 | Hejing, Bohu, Tekes, Altai, Huocheng, Artux, Kuqa, Qira, Baicheng, Korla, Moyu, Bachu, Wushi, Yengisar, Aksu, Shaya, Hotan | <0.05 |
| significant downward trend | 11 | <0 | Ulho, Mori, Qitai, Dasaka, Dushanzi, Tianshan, Xinshi, Midong, Shaybak, Toutunhe, Terrazzo | |

This table summarizes the results of county-level echinococcosis incidence trend tests, grouping counties by two trend types: significant upward trend and significant downward trend. It includes the number of counties in each group, Z value (trend direction: >0 for upward, <0 for downward), corresponding county names, and notes that a P value < 0.05 indicates statistical significance. Data were extracted from the China Infectious Disease Surveillance and Reporting System (IDSR).

Kashgar Prefecture during the early period (2005–2010). After 2017, the number of LL areas notably increased in Urumqi City and Changji Prefecture. The proportion of HH+LL clustered areas rose from 21.9% in 2005 to 44.8% in 2023, indicating a worsening spatial division.

Hotspot analysis showed that the hotspot regions were concentrated in the northern part of Xinjiang, with key areas including the Tacheng Prefecture (Tahcheng, Emin, Yumin), Bole Prefecture (Bole, Jinghe, Wenquan), and Ili Prefecture (Yining, Houcheng, Qapqal), accounting for over 60%. The number of hotspot counties increased from 12 in 2005–29 in 2023, a growth rate of 141.67%, and the distribution of hotspot areas became more concentrated. The cold spot regions

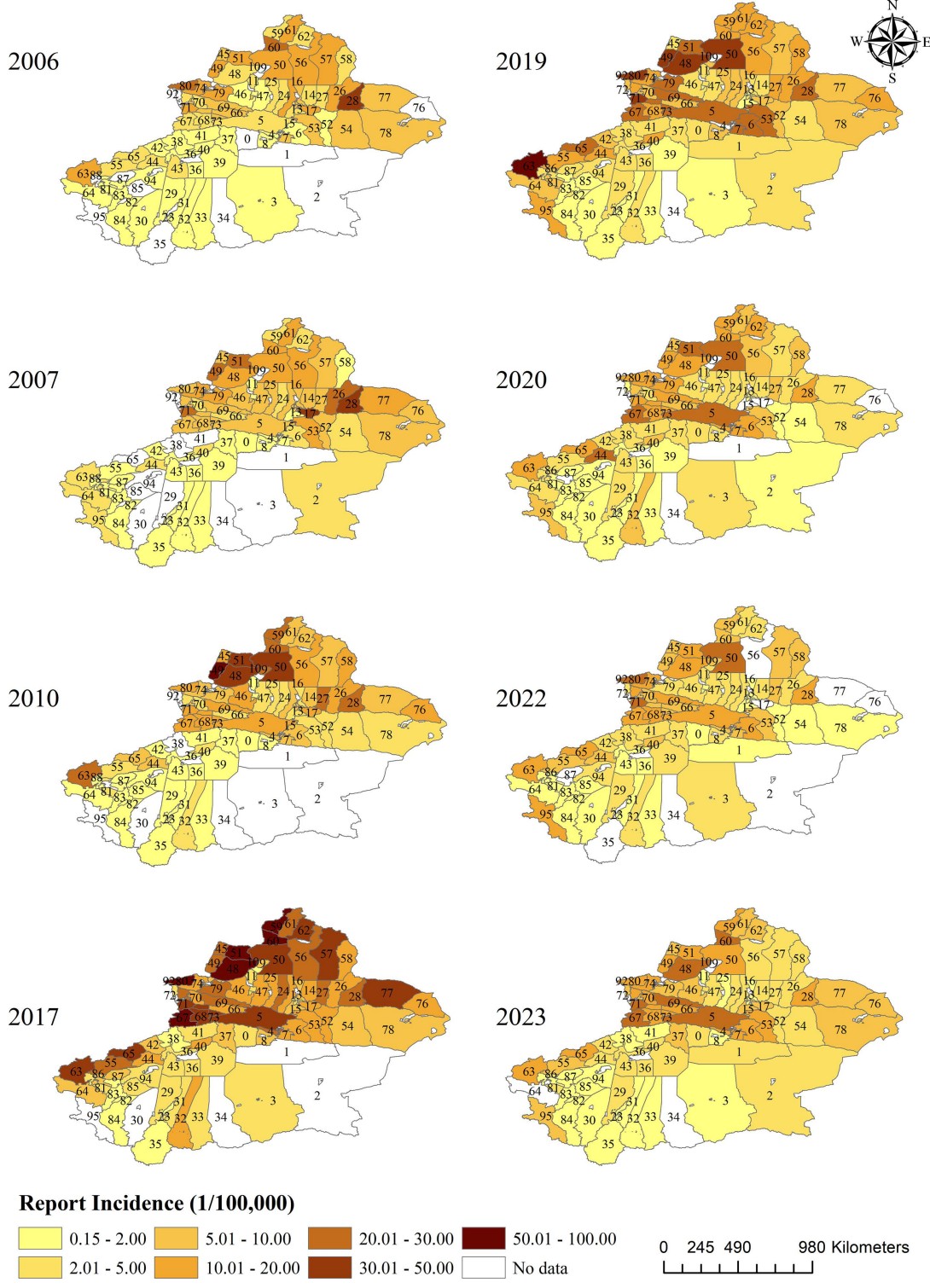

**Report Incidence (1/100,000)**

| | | |
|---|---|---|
| 0.15 - 2.00 | 5.01 - 10.00 | 20.01 - 30.00 | 50.01 - 100.00 |
| 2.01 - 5.00 | 10.01 - 20.00 | 30.01 - 50.00 | No data |

0  245  490  980 Kilometers

**Fig 3. These maps illustrate the county-level reported incidence of echinococcosis (per 100,000 population) in Xinjiang across selected years.** The color gradient represents different incidence ranges, with the legend indicating categories from 0.15–2.00 to ≥50.01. County IDs are provided in S1 Table. Base maps were sourced from the National Geographic Information Public Service Platform (www.tianditu.gov.cn, review number GS (2024) 0650) and are licensed for use under the Open Government Data License of China. Data were derived from the China Infectious Disease Surveillance and Reporting System (IDSR).

remained stable in southern Xinjiang, with the core areas being Kashgar Prefecture and Hotan Prefecture, as shown in Table 5 and Fig 4.

## Model construction and evaluation

After performing a logarithmic transformation on the echinococcosis incidence data, linear regression (LR), spatial autoregressive (SAR) model, and spatiotemporal autoregressive (STAR) model were constructed, and the results are shown in Table 5. The LR model shows that, when controlling for regional factors, the logarithmically transformed incidence rate shows an upward trend over time ($\beta = 0.028$, $P < 0.001$). The average residual level for this model was 0.061 (-0.404, 0.464), and the AIC value was 5,949.3.

SAR model built upon the LR model, takes into account spatial autocorrelation. The results show a significant positive spatial relationship between the logarithmically transformed incidence rate and the incidence rates of neighboring regions ($\rho = 0.401$, $P < 0.001$). The average residual level for this model was 0.027 (-0.873, 0.943), and the AIC value was 5,944.6, which is lower than the AIC value of the LR model (5,949.3). After accounting for spatial autocorrelation, the model fit improved.

Further, STAR model was constructed to explore the impact of spatiotemporal factors on the incidence rate of echinococcosis. The results show that spatial lag and time lag factors all have significant effects on the incidence rate. Using

**Table 5. Spatial Autocorrelation and Coldspot-Hotspot Analysis of Echinococcosis in Xinjiang, 2005-2023.**

| year | Global Autocorrelation Analysis | Local Autocorrelation Analysis | | Coldspot-Hotspot Analysis | |
| --- | --- | --- | --- | --- | --- |
| | Moran'I(Z-value, P) | High-High Clusters | Low-Low Clusters | Hotspots | Coldspots |
| 2005 | 0.183(3.858, <0.001) | 5 | 16 | 12 | 0 |
| 2006 | 0.221(5.650, <0.001) | 9 | 23 | 23 | 23 |
| 2007 | 0.236(6.477, <0.001) | 12 | 22 | 12 | 22 |
| 2008 | 0.097(2.859, <0.001) | 6 | 24 | 14 | 14 |
| 2009 | 0.218(5.987, <0.001) | 12 | 20 | 27 | 20 |
| 2010 | 0.400(6.025, <0.001) | 7 | 13 | 14 | 11 |
| 2011 | 0.189(5.076, <0.001) | 9 | 21 | 24 | 24 |
| 2012 | 0.280(4.374, <0.001) | 12 | 16 | 20 | 12 |
| 2013 | 0.251(6.889, <0.001) | 18 | 25 | 24 | 26 |
| 2014 | 0.292(4.568, <0.001) | 10 | 22 | 17 | 11 |
| 2015 | 0.219(6.080, <0.001) | 17 | 21 | 21 | 23 |
| 2016 | 0.330(5.082, <0.001) | 9 | 27 | 18 | 21 |
| 2017 | 0.230(6.325, <0.001) | 13 | 28 | 19 | 24 |
| 2018 | 0.154(4.878, <0.001) | 14 | 25 | 21 | 15 |
| 2019 | 0.142(3.858, <0.001) | 8 | 18 | 16 | 12 |
| 2020 | 0.222(6.358, <0.001) | 14 | 24 | 27 | 21 |
| 2021 | 0.149(4.697, <0.001) | 14 | 26 | 24 | 21 |
| 2022 | 0.189(5.035, <0.001) | 13 | 22 | 20 | 16 |
| 2023 | 0.251(6.436, <0.001) | 18 | 25 | 29 | 24 |

This table presents annual spatial analysis results of echinococcosis, including three parts: 1) Global Autocorrelation Analysis (Moran'I index, with Z-value and P-value in parentheses; all P < 0.001 indicate significant spatial autocorrelation); 2) Local Autocorrelation Analysis (counts of High-High clusters [high-incidence areas adjacent to high-incidence areas] and Low-Low clusters [low-incidence areas adjacent to low-incidence areas], all with P < 0.05 for statistical significance); 3) Coldspot-Hotspot Analysis (counts of hotspots [areas with significantly high incidence] and coldspots [areas with significantly low incidence], all with P < 0.05 for statistical significance). Data were extracted from the China Infectious Disease Surveillance and Reporting System (IDSR).

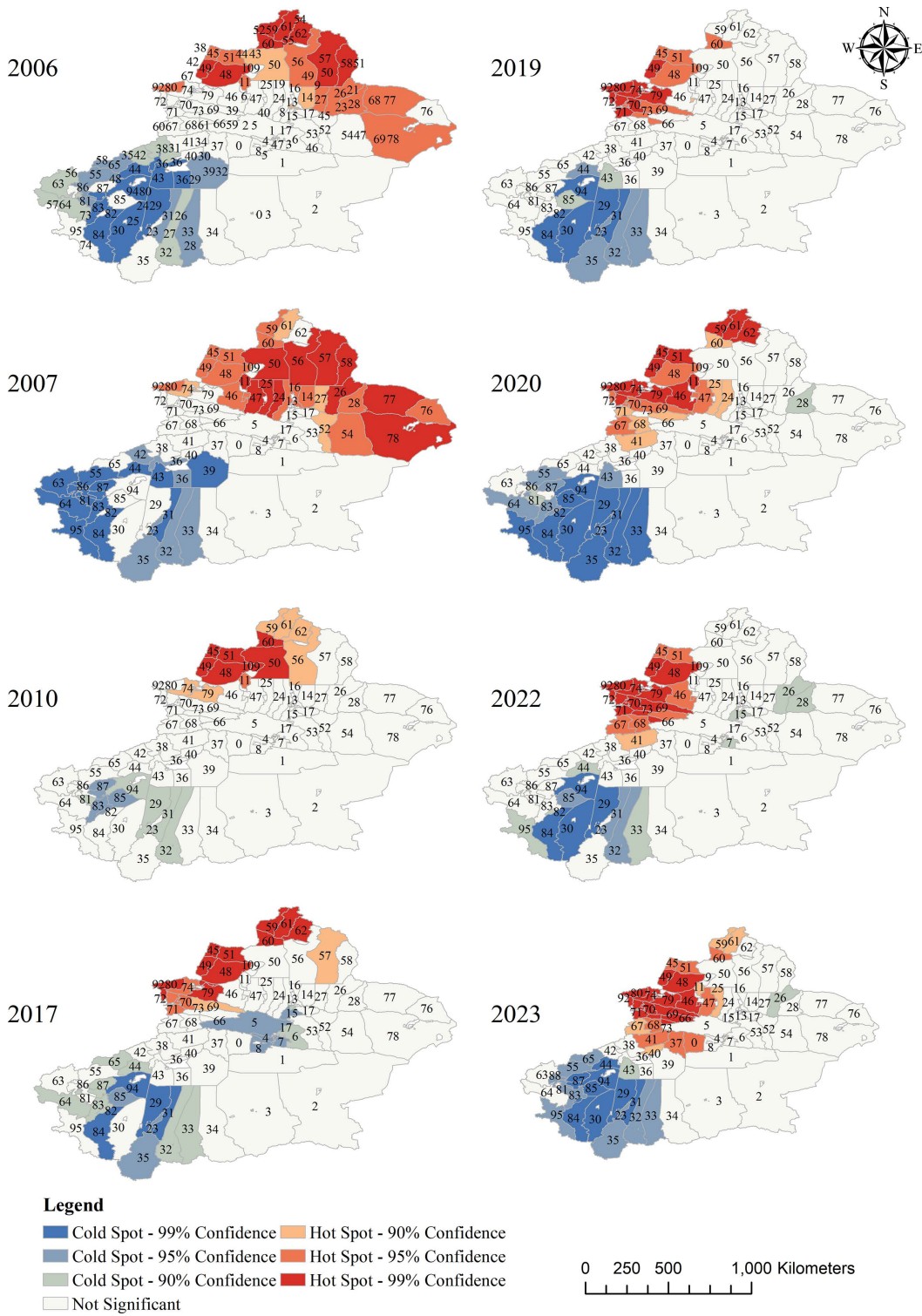

**Fig 4. Hot spots of echinococcosis reported incidence in Xinjiang, 2006, 2007, 2010, 2017, 2019, 2020, 2022, and 2023.** These maps illustrate the spatial hot spots and cold spots of echinococcosis reported incidence (per 100,000 population) at the county level in Xinjiang across selected years. The legend differentiates cold spots (blue shades) and hot spots (red shades) with varying confidence levels (90%, 95%, 99%), and gray areas indicate

**Legend**
- Cold Spot - 99% Confidence
- Cold Spot - 95% Confidence
- Cold Spot - 90% Confidence
- Not Significant
- Hot Spot - 90% Confidence
- Hot Spot - 95% Confidence
- Hot Spot - 99% Confidence

0   250  500        1,000 Kilometers

non-significant regions. County IDs are provided in S1 Table. Base maps were sourced from the National Geographic Information Public Service Platform (www.tianditu.gov.cn, review number GS (2024) 0650) and are licensed for use under the Open Government Data License of China. Data were derived from the China Infectious Disease Surveillance and Reporting System (IDSR).

2006 as the baseline, different years exhibited varying directions and intensities of impact. The period from 2011 to 2017 and 2019 showed an upward trend ($P < 0.05$), while 2020 and 2022 showed a downward trend ($P < 0.05$). The time lag coefficient was 0.420 ($P < 0.001$), indicating that the logarithmically transformed incidence rate of the previous period had a significant positive effect on the current period. The spatial lag coefficient was 0.323 ($P = 0.012$), revealing that the logarithmically transformed incidence rates of neighboring regions had a significant positive effect on the incidence rate in the current region, indicating the presence of spatial spillover effects. The average residual level of this model was 0.020 (-0.292, 0.317), and the AIC value was 2,549.4, which is lower than that of the previous two models, indicating better model fit, as shown in Table 6.

## Discussion

This study shows that from 2005 to 2023, the reported incidence rate of echinococcosis in Xinjiang fluctuated between 1.95 and 10.85 per 100,000 population, showing an increasing trend followed by a decrease, with a peak in 2017, consistent with the overall situation in China [12]. It should be noted that the reported incidence reflects only diagnosed cases registered in the surveillance system, rather than the actual rate of new infections. Given the chronic course of echinococcosis, the reported incidence is highly influenced by the extent of surveillance coverage and the intensity of active case detection.

Since 2005, echinococcosis has been included in the management of Category C notifiable infectious diseases under the Infectious Disease Prevention and Control Law, and in 2005 the national control program was launched, with central government funds transferred to support control activities in Xinjiang [13,14]. In 2006, echinococcosis was designated as

**Table 6. Model Results of Different Models.**

| Parameters | LR Model | SAR Model | STAR Model |
|---|---|---|---|
| Intercept Term | '-55.480** | '-55.682** | – |
| Coefficient | | | |
| Time | 0.028** | 0.028** | a |
| Time Lagged Term | – | – | 0.420** |
| Spatial Lagged Term | – | 0.401* | 0.323* |
| Residuals | | | |
| Min | -3.085 | -3.816 | -2.323 |
| P25 | -0.404 | -0.873 | -0.292 |
| Median | 0.061 | 0.027 | 0.020 |
| P75 | 0.464 | 0.943 | 0.317 |
| Max | 1.718 | 3.240 | 1.536 |
| AIC | 5,949.3 | 5,944.6 | 2,549.4 |

Note: ** indicates significance at the 0.01 level, and * indicates significance at the 0.05 level."a" indicates that the time coefficient was positive during 2011–2017 and 2019, negative in 2020 and 2022, with all relevant $P$ - values < 0.05.

This table compares results of three models: Logistic Regression (LR) Model, Spatial Autoregressive (SAR) Model, and Spatio-Temporal Autoregressive (STAR) Model. It includes two main sections: 1) Model Parameters (intercept term, time coefficient, time lagged term, spatial lagged term; "-" indicates the parameter is not included in the model); 2) Residual Statistics (minimum [Min], 25th percentile [P25], median, 75th percentile [P75], maximum [Max]) and Akaike Information Criterion (AIC) values. Data were derived from the China Infectious Disease Surveillance and Reporting System (IDSR).

a major infectious disease eligible for free treatment. In 2010, the "Action Plan for Echinococcosis Control (2010–2015)" was implemented, with core measures including dog registration and regular deworming, slaughterhouse inspection and safe disposal, targeted population screening and treatment, improvement of drinking water in pastoral areas, and multi-level health education [15–17]. By 2016, comprehensive control measures had been implemented all 81 endemic counties, and Xinjiang had also launched a universal health check-up program, providing annual free examinations for registered residents, which substantially improved the capacity for early case detection [18]. Therefore, from 2005 to 2017, the reported incidence increased, peaking in 2017. This rise reflected both ongoing transmission and intensified case detection driven by population-based screening, particularly among high-risk groups.

From 2017 to 2023, the reported incidence of echinococcosis in Xinjiang declined significantly, with an APC of −17.71% (95% CI: −27.81 to −10.48), indicating the positive impact of control measures [19]. During this period, the government continued to strengthen financial and technical support, achieving routine surveillance in all 81 endemic counties by 2019, promoting the construction of comprehensive intervention zones, maintaining dog management and livestock vaccination strategies, sustaining continuous health education, and implementing active population screening, all of which contributed to reducing transmission. In addition, the government raised special subsidy standards (10,000 Chinese Yuan (CNY) for cystic echinococcosis and 20,000 CNY for alveolar echinococcosis) [20,21]. By 2019, prior to the COVID-19 pandemic, the reported incidence had fallen to 6.89 per 100,000, representing a 36.6% decrease from the 2017 peak.

However, during the COVID-19 outbreak from 2020 to 2022, substantial prevention and control resources were redirected to pandemic response, limiting human and material inputs for echinococcosis control. Routine surveillance and intervention schedules were disrupted, with reduced monitoring frequency and delayed implementation of control measures in some areas, thereby weakening the effectiveness of interventions [22]. In this period, the reported incidence was 4.04, 3.65, and 3.11 per 100,000, respectively, showing a continued decline, though the decrease may partly reflect underreporting due to limited surveillance during the pandemic. In the post-pandemic era, close attention should be paid to the potential rebound of incidence. Ai J. and colleagues, using a SARIMA model, predicted a slight increase in incidence between 2022 and 2025 [23]. Our study found that in 2023, the reported incidence rebounded to 3.99 per 100,000, representing a 28.3% year-on-year increase and returning to the 2020 level. This single-year increase should be interpreted with caution, as it may partly reflect surveillance artifacts rather than a true resurgence. Nonetheless, the finding underscores the need to sustain early screening, health education, and multi-level interventions in the post-pandemic period.

The global autocorrelation analysis shows a positive spatial correlation in the reported incidence rate of echinococcosis in Xinjiang from 2005 to 2023. According to the thematic maps, counties with high incidence rates tend to cluster in the northern and central regions of Xinjiang, while counties with low incidence rates are mainly concentrated in the southern part of Xinjiang. The local spatial autocorrelation analysis further reveals the spatial heterogeneity of incidence, and the cold-hotspot analysis shows that the hotspot areas of echinococcosis in Xinjiang are concentrated in the northern region, centered around Tacheng Prefecture, Bortala Prefecture, and Ili Prefecture. Over the course of 19 years, the number of hotspot counties in Xinjiang increased from 12 to 29, with a growth rate of 141.67%, and the distribution has become more concentrated. These regions have developed livestock farming, with large numbers of livestock and frequent human-animal contact, creating favorable conditions for disease transmission [24,25]. Previous studies have shown that climate and geographical environment have a significant impact on the incidence of echinococcosis. Cold and humid conditions are favorable for the survival, development, and reproduction of echinococcosis eggs. For every 1 mm increase in average summer precipitation, the risk of echinococcosis increases by 0.60% [2,26]. The northern part of Xinjiang has a temperate continental climate with relatively high precipitation and abundant pastures, providing an environment conducive to the survival and reproduction of parasites and their hosts [27]. The southern part of Xinjiang has a warm temperate continental arid climate, with little rainfall throughout the year. It is affected by sandstorms from the Taklamakan Desert and the

Kunlun Mountains, with frequent windy and sandy weather that restricts the survival and transmission of parasites [2,28]. As a result, the incidence rate is relatively low, and the cold spot areas remain stable in the southern Xinjiang region, centered around Kashgar Prefecture and Hotan Prefecture. However, cold spot areas are not risk-free. With changes in environmental and socio-economic conditions, outbreaks may also occur in these regions.

This study developed LR, SAR, and STAR models. Among the three models, the STAR model had the lowest AIC value and the lowest average residual level, indicating that the STAR model provided the best fit.This indicates that when analyzing the incidence of echinococcosis, considering both time-lag and spatial-lag variables can more comprehensively reflect the patterns of incidence changes. The results of the STAR model show that year, spatial lag, and time lag factors all have significant effects on incidence ($P<0.05$). The incidence of echinococcosis in Xinjiang exhibits temporal continuity and spatial spillover effects. Therefore, a dynamic prevention and control system should be established, with flexible adjustments of control measures according to epidemic changes, and continuous monitoring of historically high-incidence areas to prevent a resurgence of the epidemic. Considering spatial spillover effects, prevention and control efforts should not only focus on local epidemics but also strengthen monitoring and control of surrounding areas, adopting joint prevention and control measures to prevent the spread of the disease between regions [29].

This study has several limitations. First, the early monitoring system in Xinjiang was incomplete, resulting in missing incidence data for certain years and counties, which affects the continuity of trend analysis and the completeness of spatial distribution research. Second, the reported incidence data reflect only diagnosed and registered cases, which may differ from the true infection rate due to underreporting or variations in surveillance coverage and active case-finding intensity. Third, the study lacks quantitative data on the implementation of control interventions (e.g., dog deworming, livestock vaccination, and active screening campaigns), which limits the ability to directly evaluate the impact of these measures on incidence trends. Fourth, this study mainly considers temporal and spatial factors, without incorporating climate change, socio-economic factors, and other potential determinants, which may limit a comprehensive understanding of the drivers of incidence changes.

Given these limitations, future research could incorporate climate and socio-economic variables, obtain quantitative intervention data [30], and conduct county-level comparative studies (e.g., active screening vs. non-active screening areas) to more systematically assess the effectiveness of control measures and further elucidate the epidemiological patterns of echinococcosis in Xinjiang.

## Supporting information

**S1 Table. County ID, names, and median reported incidence of echinococcosis in Xinjiang, 2005–2023 (per 100,000 population).** County IDs correspond to those used in all figures and tables. Data were obtained from the China Infectious Disease Surveillance and Reporting System (IDSR) and Xinjiang CDC records.
(DOCX)

**S1 Fig. Median reported incidence of echinococcosis in Xinjiang by county, 2005–2023 (per 100,000 population).** County IDs correspond to S1 Table. Data source: IDSR and Xinjiang CDC records. Base maps were sourced from the National Geographic Information Public Service Platform (www.tianditu.gov.cn, review number GS (2024) 0650) and are licensed for use under the Open Government Data License of China. Data were derived from the China Infectious Disease Surveillance and Reporting System (IDSR).
(TIF)

## Author contributions

**Conceptualization:** Zhe Yin, Adili Simayi.

**Data curation:** Zhe Yin, Guangzhong Shi, Yalikun Maimaitiyiming, Qi Wang, Kaisaer Tuerxunjiang.

**Formal analysis:** Zhe Yin, Guangzhong Shi.

**Investigation:** Yalikun Maimaitiyiming, Qi Wang, Kaisaer Tuerxunjiang.

**Methodology:** Zhe Yin.

**Project administration:** Guangzhong Shi.

**Resources:** Jiangshan Zhao.

**Writing – original draft:** Zhe Yin.

**Writing – review & editing:** Adili Simayi, Jiangshan Zhao.

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
