## [Decision Letter · Decision Letter 0]

4 Aug 2025

Spatiotemporal Distribution and Control of Echinococcosis in Xinjiang, 2005-2023

Dear Dr. Zhao,

Thank you for submitting your manuscript to PLOS Neglected Tropical Diseases. After careful consideration, we feel that it has merit but does not fully meet PLOS Neglected Tropical Diseases's publication criteria as it currently stands. Therefore, we invite you to submit a revised version of the manuscript that addresses the points raised during the review process.

Please submit your revised manuscript within 60 days Oct 03 2025 11:59PM. If you will need more time than this to complete your revisions, please reply to this message or contact the journal office at plosntds@plos.org. Please include the following items when submitting your revised manuscript:

We look forward to receiving your revised manuscript.

Kind regards,

Majid Fasihi Harandi, PhD

Academic Editor

Uriel Koziol

Section Editor

Shaden Kamhawi

co-Editor-in-Chief

Paul Brindley

co-Editor-in-Chief

**Journal Requirements:**

At this stage, the following Authors/Authors require contributions: Zhe Yin, Guangzhong Shi, Yalikun Maimaitiyiming, Qi Wang, Kaisaer Tuerxunjiang, Adili Simayi, and Jiangshan Zhao. Please ensure that the full contributions of each author are acknowledged in the "Add/Edit/Remove Authors" section of our submission form.

3) Tables should not be uploaded as individual files. Please remove these files and include the Tables in your manuscript file as editable, cell-based objects. For more information about how to format tables, see our guidelines:

https://journals.plos.org/plosntds/s/tables 

4) We note that your Data Availability Statement is currently as follows: "The datasets generated and/or analyzed during the current study are not publicly available due to the confidentiality and sensitivity of county-level data across the Xinjiang region, as well as our institution's strict data management policies.". Please confirm at this time whether or not your submission contains all raw data required to replicate the results of your study. Authors must share the “minimal data set” for their submission. PLOS defines the minimal data set to consist of the data required to replicate all study findings reported in the article, as well as related metadata and methods (https://journals.plos.org/plosone/s/data-availability#loc-minimal-data-set-definition).

- The points extracted from images for analysis..

5) Please ensure that the funders and grant numbers match between the Financial Disclosure field and the Funding Information tab in your submission form. Note that the funders must be provided in the same order in both places as well.

**Reviewers' Comments:**

Reviewer's Responses to Questions

**Key Review Criteria Required for Acceptance?**

**Methods:**

-Are the objectives of the study clearly articulated with a clear testable hypothesis stated?

-Is the study design appropriate to address the stated objectives?

-Is the population clearly described and appropriate for the hypothesis being tested?

-Is the sample size sufficient to ensure adequate power to address the hypothesis being tested?

-Were correct statistical analysis used to support conclusions?

-Are there concerns about ethical or regulatory requirements being met?

Reviewer #1: (No Response)

Reviewer #2: Spatiotemporal models are powerful tools especially for analyzing and predicting the trend of echinococcosis given that its transmission is so complex. However, the data factors in this study are so simple including human cases in different areas and different time points. The simple size was OK for the analysis. The authors should provide details of control measures.

Reviewer #3: It's always valuable to have new information from one of the most endemic areas in the world.

In the background, I suggest not mentioning mortality in sheep and goats. It's uncertain, and animals are usually slaughtered before the cysts severely damage their health. CE is a public health problem, and this issue should be emphasized. In livestock, it could only be of interest if there are CE confiscations in slaughterhouses. There is data!

Another issue is that a long period of time is taken, but there's no mention of whether there are official control programs or what they consist of. According to the last paragraph of background, there is no program, and I don't believe that's the case. Please explain this, as it will be vital to interpreting the data found.

The statistical analysis is avoiding the use of extensive information and speaks well of the data analysis.

Regarding the results, I personally think it's better to present the exact P values.

Again, the lack of information on programs and actions makes it difficult to know if the fluctuations are real. This is a problem with reporting, active case-finding campaigns, or the result of deworming dogs or vaccinating sheep.

Discussion. Finally, program actions and case-finding for asymptomatic cases are described. Therefore, the trends are clearly influenced by this.

In the graphs and throughout the work, avoid using hydatidosis and replace it with cystic echinococcosis.

However, it's already too late.

We suggest reorganizing the work, perhaps describing the control actions in the introduction. And, possibly, a timeline graphing the incidence by year and identifying the control or search actions initiated.

Clearly, the objective is poorly stated, as it proposes to advise authorities who are already working. It would be appropriate to change it to something like helping authorities evaluate the progress of actions. And these actions should be analyzed in the discussion.

**Results:**

-Does the analysis presented match the analysis plan?

-Are the results clearly and completely presented?

-Are the figures (Tables, Images) of sufficient quality for clarity?

Reviewer #1: (No Response)

Reviewer #2: As the data are not clearly classified and the results are basically not supporting the trend of disease endemic and control results.

Reviewer #3: see method

**Conclusions:**

-Are the conclusions supported by the data presented?

-Are the limitations of analysis clearly described?

-Do the authors discuss how these data can be helpful to advance our understanding of the topic under study?

-Is public health relevance addressed?

Reviewer #1: (No Response)

Reviewer #2: The conclusions are not fully supported by the data.

Reviewer #3: see method

**Editorial and Data Presentation Modifications?**

Reviewer #1: (No Response)

Reviewer #2: In this study entitled “Spatiotemporal Distribution and Control of Echinococcosis in Xinjiang, 2005-2023,” Zhe Yin et al. collected annual human echinococcosis cases from each county of Xinjiang Uygur Autonomous Region (province level) to fit the spatiotemporal model, showing the distribution of echinococcosis in both spatial and temporal dimensions. They also used the annual case reports from each county to determine the control progress from 2005 to 2023. Given that echinococcosis is an important zoonosis and very few control study in recent years, the data is very crucial if the authors give more details on the control measures used in the control programme. Further more, in this study, the authors showed that after the control program was implemented, the number of human cases increased, peaking in 2017 at 10.8/1,000,000. The authors suggested that this might be due to the active population screening procedure. Additionally, they mentioned that the control program was not implemented evenly or equally in these counties in Xinjiang from 2005 to 2017. I believe these data are useful for demonstrating that active population screening may cause an increase in reported human cases. Therefore, the authors may design a study comparing these counties with active screening to those without screening. It is not appropriate to mix these data together to show an “increasing” or “decreasing” trend. After 2017, all the targeted counties were implementing the control program, the authors should provide details about the control measures used and how the program was monitored. Furthermore, if they claim that the incidence from the active screening procedure can be used to profile the control program, they should provide more details about the population screening, such as whether it was conducted on the entire population.

Spatiotemporal models are good and powerful models to analyze the spread of infectious diseases. In terms of zoonosis, the data should not only include the time and location of cases and incidence rates, but also factors such as health care capacity, livestock husbandry, agricultural activities, rural populations, and education levels, which are likely to play significant roles in determining the prevalence of CE. Most importantly, the model should be used to describe the transmission pathways of this helminth zoonosis across different regions over time, to identify high-risk areas and periods, and most importantly, to forecast potential future trends of the epidemic to support public health and design control measures.

Minors

1. Abstract results: “The reported incidence of echinococcosis in Xinjiang ranged from 1.95 to 10.85 per 100,000.” In which years did these figures occur? Does this mean the incidence increased after the control program was implemented?

2. Abstract: The authors should present some specific data in the abstract results, rather than just using terms like “low,” “hotspot,” or “cold spots.”

3. What does “spatial lag and temporal lag significantly affecting incidence rates” mean? Please provide data to support these results!

4. Abstract Conclusion: The conclusion should clearly summarize the epidemiological situation and control efficacy, aligning with the title and aims of the study.

5. Lines 71-73: Change “The incidence data of echinococcosis reported from 2005 to 2023 in Xinjiang and 96 districts/counties were obtained from the China Infectious Disease Surveillance and Reporting System” to “Data on the incidence of echinococcosis reported in Xinjiang Uygur Autonomous Region and its 96 districts/counties during 2005–2023 were obtained from the China Infectious Disease Surveillance and Reporting System (IDSR).”

Results/Discussion

6. The authors indicate that the control program was initiated in 2005 with an incidence of 1.95/100,000, and then the incidence gradually increased to 10.8/100,000. The authors should discuss the reasons for this increase. Was the control program ineffective? Or were there other reasons for the rise? After 2017, the incidence decreased and remained at 3.99/100,000 in 2023, which is still higher than when the control program began. I have not seen a discussion of these results.

7. Line 298-300: “The subsidy for cystic echinococcosis has increased to 10,000, while the subsidy for alveolar echinococcosis has reached 20,000.” Suggestion: Specify the currency to avoid ambiguity.

Reviewer #3: (No Response)

**Summary and General Comments:**

Reviewer #1: Review Report for Manuscript PNTD-D-24-56789

Dear Dr. Majid Fasihi Harandi and Editorial Team,

Thank you for inviting me to review the manuscript titled "Spatiotemporal Distribution and Control of Echinococcosis in Xinjiang, 2005-2023". The study provides valuable insights into the epidemiological dynamics of echinococcosis in Xinjiang over nearly two decades, which is critical for understanding and managing this neglected tropical disease in Central Asia. Below are my comments and suggestions to improve clarity, accuracy, and impact:

**Major Comments:**

Use of the term "outbreak" (Line 194)

The authors describe a "severe initial outbreak" in Ulho County in 2005. However, echinococcosis is a chronic parasitic disease with insidious progression, which does not align with the classical epidemiological definition of an "outbreak" (typically acute and rapid transmission). Furthermore, since the study period starts in 2005, the term "initial" lacks context without pre-2005 data.

Recommendation: Replace "outbreak" with "high case detection" or similar phrasing. Provide historical context for temporal claims or clarify that "initial" refers to the study period rather than disease emergence.

Interpretation of 2023 incidence increase (Lines 311–313)

The 28.30% year-on-year rise in 2023 is interpreted as a "clear upward trend," but this conclusion relies on a single data point without post-2023 validation. Such fluctuations could reflect surveillance artifacts (e.g., post-pandemic catch-up screenings) rather than true resurgence.

Recommendation: Acknowledge the limitation of interpreting short-term fluctuations and discuss alternative explanations (e.g., surveillance intensity).

Terminology: Echinococcus vs. echinococcosis (Figures/Tables)

The manuscript inconsistently uses Echinococcus (the parasite genus) where echinococcosis (the disease) is appropriate (e.g., in figures/tables). Additionally, Echinococcus should be italicized as a genus name.

Recommendation: Replace Echinococcus with "echinococcosis" or "Echinococcus infection" in all non-taxonomic contexts. Italicize Echinococcus when referring to the genus.

**Minor Comments:**

Units for subsidy funds (Line 299)

The currency unit for subsidies is unspecified.

Recommendation: State whether amounts are in CNY (RMB) or USD.

Redundancy in Table 5 and Figure 4

The spatial regression results are duplicated in Table 5 and Figure 4.

Recommendation: Retain the table in the main text and move the figure to supplementary materials (or vice versa).

Map visualization improvements (Figures 3 and 4)

County labels are overcrowded, hindering readability.

Recommendation: Use numbered codes for counties, with a legend highlighting high/low-incidence areas (e.g., hotspots in red).

Year selection rationale in the Maps (Figures 3 and 4)

The rationale for selecting specific years (e.g., 2005, 2010, 2015, 2020) is unclear.

Recommendation: Justify year selection (e.g., alignment with policy changes, surveillance milestones).

Supplementary spatial analysis

Cumulative annual incidence or average annual incidence maps could better illustrate long-term spatiotemporal patterns. Recommendation: Add such maps as supplementary figures.

Additional evaluation opinion of "incidence rate" terminology

The manuscript uses "incidence rate" throughout, but echinococcosis is a chronic condition where detected cases often reflect long-term infections rather than new acute infections. As noted in the Discussion, case identification may correlate with screening programs rather than true incidence. Recommendation: Clarify in the Methods/Discussion that "incidence" here refers to reported cases (not true disease onset) and discuss screening-driven biases.

This study addresses a critical public health issue with robust spatiotemporal analyses. Addressing the above points will strengthen the manuscript’s scientific rigor and clarity. I recommend minor revisions prior to publication.

Thank you for the opportunity to review this work.

Best regards,

Xu Wang, PhD

National Institute of Parasitic Diseases, China CDC

wangxu@nipd.chinacdc.cn

Reviewer #2: In deed, echinococcosis needs new "trend forecasting methods" and new control methods.

Reviewer #3: (No Response)

PLOS authors have the option to publish the peer review history of their article (what does this mean? ). If published, this will include your full peer review and any attached files.

**Do you want your identity to be public for this peer review?** For information about this choice, including consent withdrawal, please see our Privacy Policy .

Reviewer #1: No

Reviewer #2: No

Reviewer #3: **Yes: ** edmundo larrieu

**Figure resubmission:**

**Reproducibility:**



---

## [Decision Letter · Decision Letter 1]

23 Oct 2025

Thank you for submitting your manuscript to PLOS Neglected Tropical Diseases. After careful consideration, we feel that it has merit but does not fully meet PLOS Neglected Tropical Diseases's publication criteria as it currently stands. Therefore, we invite you to submit a revised version of the manuscript that addresses the points raised during the review process.

Response to Reviewers
Revised Manuscript with Track Changes
Manuscript

We look forward to receiving your revised manuscript.

Shaden Kamhawi

co-Editor-in-Chief

Paul Brindley

co-Editor-in-Chief

**Additional Editor Comments :**

Please revise your manuscript in accordance with the reviewer’s comments, giving special attention to each of the issues raised by Reviewer #3. In particular, the reviewer has emphasized the need to clearly state the study objectives before the methods, and to include a concise summary of the control strategies implemented during the study period such as dog deworming, sheep vaccination, and ultrasound-based case detection, indicating when and where these activities were applied.

The reviewer also requests clarification of whether the surveillance system distinguishes new cases from re-operations or re-admissions, and whether incidence calculations excluded such cases. Finally, please ensure that the discussion more clearly links the described control measures with the observed epidemiological trends, so that the impact of these actions is coherently interpreted in light of the study’s objectives.

**Journal Requirements:**

1) Please upload figure 1 as a separate Figure file in .tif or .eps format. For more information about how to convert and format your figure files please see our guidelines: 

2) We have noticed that you have uploaded Supporting Information files, but you have not included a complete list of legends. Please add a full list of legends for all your Supporting Information files after the references list.

3) We notice that your supplementary figure and table are uploaded with the file types 'Figure' and 'Table'. Please amend the file types to 'Supporting Information'. Please ensure that each Supporting Information file has a legend listed in the manuscript after the references list.

4)  Figures 3, 4, and S1. Please ensure to include in the figures legends direct link(s) to the base layer of the map(s) and link(s) to the terms of use / license information for the base layer image or shapefile. 

5) We note that you have indicated that there are restrictions to data sharing for this study. PLOS only allows data to be available upon request if there are legal or ethical restrictions on sharing data publicly. For more information on unacceptable data access restrictions, please see https://journals.plos.org/plosntds/s/data-availability#loc-unacceptable-data-access-restrictions.

Please confirm whether all data needed to replicate the study's findings are in the submission and/or supplementary materials. If so, please update your Data Availability statement to reflect the information you provide.

**Reviewers' comments:**

**Key Review Criteria Required for Acceptance?**

**Methods**

-Are the objectives of the study clearly articulated with a clear testable hypothesis stated?

-Is the study design appropriate to address the stated objectives?

-Is the population clearly described and appropriate for the hypothesis being tested?

-Is the sample size sufficient to ensure adequate power to address the hypothesis being tested?

-Were correct statistical analysis used to support conclusions?

-Are there concerns about ethical or regulatory requirements being met?

Reviewer #2: The authors have changed according to the reviewer's comments and suggestions.

Reviewer #3: I believe it would help better interpret the work if the objectives of the study were clearly stated before the methods.

Another issue, necessary for subsequent data analysis, is a summary presentation of the control strategies applied during the study period. For example, are dogs regularly dewormed? Are sheep vaccinated? Since when? Are there massive active search programs using ultrasound? Do these activities cover the entire region or are they limited to certain areas?

Method: Does the surveillance system distinguish new cases from reoperations or readmissions? Is the incidence calculated by excluding reoperations or readmissions?

Discussion: The association with control appears here, but it is confusing. Please describe the most important aspects as suggested in the introduction, and the discussion should clarify the impact of these actions on the trend.

Excellent statistical analysis

**Results**

-Does the analysis presented match the analysis plan?

-Are the results clearly and completely presented?

-Are the figures (Tables, Images) of sufficient quality for clarity?

Reviewer #2: the results are clearly presented and discussed.

Reviewer #3: (No Response)

**Conclusions**

-Are the conclusions supported by the data presented?

-Are the limitations of analysis clearly described?

-Do the authors discuss how these data can be helpful to advance our understanding of the topic under study?

-Is public health relevance addressed?

Reviewer #2: yes, the conclusions are supported bu the presented data.

Reviewer #3: Can be improved?

**Editorial and Data Presentation Modifications?**

Reviewer #2: still some spelling errors need to be carefully checked and changed, such as "Echinococcusis or echinococcusis" should be "Echinococcosis or echinococcosis". I wish to accept for publication.

Reviewer #3: (No Response)

**Summary and General Comments**

Reviewer #2: the revised MS has been changed and corrected according to the reviewer's comments and suggestions. I suggest accept it for publication.

Reviewer #3: (No Response)

PLOS authors have the option to publish the peer review history of their article (what does this mean? ). If published, this will include your full peer review and any attached files.

**Do you want your identity to be public for this peer review?** For information about this choice, including consent withdrawal, please see our Privacy Policy .

Reviewer #2: No

**Figure resubmission:****Reproducibility:** To enhance the reproducibility of your results, we recommend that authors of applicable studies deposit laboratory protocols in protocols.io, where a protocol can be assigned its own identifier (DOI) such that it can be cited independently in the future. Additionally, PLOS ONE offers an option to publish peer-reviewed clinical study protocols. Read more information on sharing protocols at https://plos.org/protocols?utm_medium=editorial-email&utm_source=authorletters&utm_campaign=protocols

---

## [Editor Report · Decision Letter 2]

19 Nov 2025

Dear Zhao,

We are pleased to inform you that your manuscript 'Spatiotemporal Distribution and Control of Echinococcosis in Xinjiang, 2005-2023' has been provisionally accepted for publication in PLOS Neglected Tropical Diseases.

Best regards,

Majid Fasihi Harandi, PhD

Academic Editor

Uriel Koziol

Section Editor

Shaden Kamhawi

co-Editor-in-Chief

Paul Brindley

co-Editor-in-Chief

---

## [Editor Report · Acceptance letter]

Dear Zhao,

We are delighted to inform you that your manuscript, "Spatiotemporal Distribution and Control of Echinococcosis in Xinjiang, 2005-2023," has been formally accepted for publication in PLOS Neglected Tropical Diseases.

Best regards,

Shaden Kamhawi

co-Editor-in-Chief

Paul Brindley

co-Editor-in-Chief
